**Data Availability Statement:** All relevant data are within the manuscript and its Supporting Information files.

# Opportunistic screening for atrial fibrillation by clinical pharmacists in UK general practice during the influenza vaccination season: A cross-sectional feasibility study

**Vilius Savickas**[1], **Adrian J. Stewart**[2], **Melanie Rees-Roberts**[3], **Vanessa Short**[3,4], **Sukvinder K. Bhamra**[1], **Sarah A. Corlett**[1], **Alistair Mathie**[1], **Emma L. Veale**[1]*

1 Medway School of Pharmacy, University of Kent and University of Greenwich, Chatham Maritime, Kent, United Kingdom, 2 Medway Maritime Hospital, Cardiology, Gillingham, Kent, United Kingdom, 3 Centre for Health Services Studies, University of Kent, Canterbury, United Kingdom, 4 Newton Place Surgery, Faversham, United Kingdom

* e.l.veale@kent.ac.uk

## Abstract

### Background

Growing prevalence of atrial fibrillation (AF) in the ageing population and its associated life-changing health and resource implications have led to a need to improve its early detection. Primary care is an ideal place to screen for AF; however, this is limited by shortages in general practitioner (GP) resources. Recent increases in the number of clinical pharmacists within primary care makes them ideally placed to conduct AF screening. This study aimed to determine the feasibility of GP practice–based clinical pharmacists to screen the over-65s for AF, using digital technology and pulse palpation during the influenza vaccination season.

### Methods and findings

Screening was conducted over two influenza vaccination seasons, 2017–2018 and 2018–2019, in four GP practices in Kent, United Kingdom. Pharmacists were trained by a cardiologist to pulse palpate, record, and interpret a single-lead ECG ($_{sL}$ECG). Eligible persons aged ≥65 years (y) attending an influenza vaccination clinic were offered a free heart rhythm check. Six hundred four participants were screened (median age 73 y, 42.7% male). Total prevalence of AF was 4.3%. All participants with AF qualified for anticoagulation and were more likely to be male (57.7%); be older; have an increased body mass index (BMI); and have a CHA$_2$DS$_2$-VASc (Congestive heart failure, Hypertension, Age ≥ 75 years, Diabetes, previous Stroke, Vascular disease, Age 65–74 years, Sex category) score ≥ 3. The sensitivity and specificity of clinical pharmacists diagnosing AF using pulse palpation was 76.9% (95% confidence interval [CI] 56.4–91.0) and 92.2% (95% CI 89.7–94.3), respectively. This rose to 88.5% (95% CI 69.9–97.6) and 97.2% (95% CI 95.5–98.4) with an $_{sL}$ECG. At follow-up, four participants (0.7%) were diagnosed with new AF and three (0.5%) were initiated on anticoagulation. Screening with $_{sL}$ECG also helped identify new non-AF cardiovascular diagnoses, such as left ventricular hypertrophy, in 28 participants (4.6%). The screening

**Funding:** ELV, AM, SKB, and SAC were awarded a medical education grant (MEGs) from Bayer UK (UKBAY09170342a, https://www.bayer.co.uk/). ELV was awarded a grant from Kent Surrey and Sussex Community Education Providers Network (CEPN) for pharmacist training. The funders had no role in study design, data collection and analysis, decision to publish, or preparation of the manuscript.

**Competing interests:** The authors have declared that no competing interests exist.

**Abbreviations:** $_{12L}$ECG, 12-lead ECG; AEB, atrial ectopic beats; AF, atrial fibrillation; AVB, atrioventricular block; BBB, bundle branch block; BMI, body mass index; bpm, beats per minute; CHA$_2$DS$_2$-VASc, Congestive heart failure, Hypertension, Age $\geq$ 75 years, Diabetes, previous Stroke, Vascular disease, Age 65–74 years, Sex category; CI, confidence interval; COPD, chronic obstructive pulmonary disease; ESC, European Society of Cardiology; FDR, false-discovery rate; GP, general practitioner; HCP, healthcare professional; KMD, Kardia Mobile Device; NICE, National Institute for Health and Care Excellence; OAC, oral anticoagulant; PAF, paroxysmal AF; PDAF, Pharmacists Detecting Atrial Fibrillation; PIL, participant information leaflet; PIPS, Public Involvement in Pharmacy Studies; PPV, positive predictive value; QALY, quality-adjusted life year; $_{SL}$ECG, single-lead ECG; SR, sinus rhythm; STARD, Standards for Reporting Diagnostic accuracy studies.

strategy was cost-effective in 71.8% and 64.3% of the estimates for $_{SL}$ECG or pulse palpation, respectively. Feedback from participants (422/604) was generally positive. Key limitations of the study were that the intervention did not reach individuals who did not attend the practice for an influenza vaccination and there was a limited representation of UK ethnic minority groups in the study cohort.

## Conclusions

This study demonstrates that AF screening performed by GP practice–based pharmacists was feasible, economically viable, and positively endorsed by participants. Furthermore, diagnosis of AF by the clinical pharmacist using an $_{SL}$ECG was more sensitive and more specific than the use of pulse palpation alone. Future research should explore the key barriers preventing the adoption of national screening programmes.

## Author summary

### Why was this study done?

➢ Atrial fibrillation (AF), which is often symptomless, is associated with an increased risk of developing stroke or heart failure. The prevalence of AF increases with age. Integration of screening programmes alongside existing healthcare services and infrastructure, utilising trained healthcare professionals (HCPs), must be sustainable.

➢ Screening for AF at influenza vaccination clinics using clinical pharmacists may be cost-effective and target a relevant, at-risk proportion of the population (e.g., $\geq$65 y, with multiple conditions).

### What did the researchers do and find?

➢ Using a single-time-point screening strategy that selectively targeted 604 people $\geq$65 y old, attending influenza vaccination clinics at participating general practitioner practices, we showed that appropriately trained clinical pharmacists could screen and detect AF.

➢ A participant experience questionnaire showed, generally, that participants were highly satisfied with their consultation and thought AF screening was important.

➢ We found that screening for AF during the influenza vaccination season, using clinical pharmacists and automated digital technology, was more reliable and cost-effective than pulse palpation alone.

### What do these findings mean?

➢ This work demonstrates a feasible approach to annual AF screening in primary care by clinical pharmacists using digital technology that could be readily adopted by

general practices, delivering annual influenza vaccinations to the over-65s and adapted to involve other HCPs.

➢ Further studies are needed to investigate how to broaden AF screening to those at risk who do not participate in the influenza vaccination and to explore the key barriers outlined by policy makers, which have delayed the adoption of a national AF screening programme.

## Introduction

Routine screening for atrial fibrillation (AF) is currently not endorsed by the UK National Screening Committee [1]. The growing prevalence of AF [2,3] and its associated life-changing health implications [4,5], combined with the impact of AF on national health resources [6] that can occur as a result of the disease not being detected early, have led to a growing medical consensus, backed by public health policy, to improve the early detection and treatment of AF [7–9].

The prevalence and severity of AF increases with age [10], and the older-aged population are most at risk of experiencing an AF-related stroke and/or heart failure [4,5]. Furthermore, the risk of the disease has been shown to be exacerbated when associated with other co- or multimorbidities, such as hypertension and heart failure [5,11–14]. For persons aged 55 y or older, the lifetime risk of developing AF increases from one in five to one in three in the presence of one or more morbidities [11]. The proportion of over-65s experiencing two or more chronic conditions is 54%, increasing to 69% for those over the age of 85 [15]. Thus, the older, ageing population remains key to any future national screening plans, as highlighted by the European Society of Cardiology (ESC) guidelines [8]. When, where, and how this population is targeted remains a key consideration to any future screening initiatives, in order to maximise socioeconomic outcomes.

Primary care is considered to be central to improving the early detection of AF, as this is where the majority of the populations' health is routinely managed. In England, general practitioner (GP) surgeries provide over 300 million patient consultations a year [16], making this location ideal for health screening [17–21]. The chronic shortage of doctors and nurses in the UK [22], and elsewhere, impacts heavily on patient access to primary care [23], despite efforts to retain and increase GP numbers [24]. To address this issue, NHS England have pledged to fund an additional 20,000 healthcare professionals (HCPs) by 2023/2024 to support GPs [25], with initial funding targeted to social prescribers and pharmacists. This builds on previous NHS investment into the 'Clinical Pharmacists in General Practice' pilot scheme, which, since 2015, has recruited over 1,000 full-time clinical pharmacists [26]. As such, healthcare interventions such as AF screening are likely to be delivered by another HCP, other than a GP. Clinical pharmacists are well placed to apply their in-depth knowledge of medicines, toxicology, pharmacokinetics, and therapeutics to deliver patient-centred care that promotes health, wellbeing, and disease prevention, in all patient-care settings [27,28]. The development of newer and better screening methods for AF are also being shown to improve the detection of AF and are helping to overcome some of the limitations and barriers experienced using older, more conventional methods [29,30].

In this 'Pharmacists Detecting Atrial Fibrillation' (PDAF) study, we aimed to determine the feasibility of general practice–based clinical pharmacists screening the over-65s for AF, using

digital technology and a single-time-point screening strategy combined with another annual healthcare intervention, the influenza vaccination. We evaluated the use of a single-lead electrocardiogram ($_{SL}$ECG) device compared with pulse palpation alone, as the latter is a current recommendation for AF detection [31], and the economic impact of both methods particularly in relation to false-discovery rates (FDRs). Finally, we sought feedback from the participants about the service that was provided. A preliminary account of some of these data has been reported previously [32].

## Methods

This study is reported as per the Standards for Reporting Diagnostic accuracy studies (STARD) checklist (S1 STARD Checklist).

### Study design

A single-time-point screening strategy was used to detect AF in patients aged 65 y or over attending the annual influenza vaccination at their GP practice, using clinical pharmacists to conduct the screening. Screening was conducted over two influenza vaccination seasons, from 28 October 2017 to 22 February 2018 and then from 2 October to 14 December 2018. The study protocol design was described in a previous publication [33]. In brief, five clinical pharmacists were recruited from Kent Community NHS Foundation Trust, another pharmacist was already embedded in a participating practice, and another was provided by the Medway School of Pharmacy, University of Greenwich and Kent. All pharmacists received training before and during the study to implement the screening protocol. Four GP practices across the NHS Canterbury and Coastal Clinical Commissioning Group participated in the study. Patients aged 65 or over attending an influenza clinic at a participating practice were eligible to have the rate and rhythm of their heart assessed, using pulse palpation and an $_{SL}$ECG device (AliveCor Kardia Mobile Device [KMD]). Exclusions from screening included anyone with a pacemaker, those with a severe coexisting medical condition (e.g., cancer with <1 month (mo) of life expectancy), or those who were not able to provide informed consent at time of screening because of a lack of mental capacity. Patients with preexisting AF were not excluded, as it was assumed that most participants with AF would self-exclude, whereas those that did not would act as positive test controls. Pharmacists were unaware of preexisting AF diagnoses in participants prior to screening. Participants could be screened at the clinic or could opt for a prebooked appointment. Screening was advertised via posters, leaflets, text messages, staff, clinical pharmacists, or a member of the study team. Participants were recruited using a consecutive sampling approach, meaning that any participants attending influenza vaccination clinics at participating practices during the studied time periods and fulfilling the study inclusion criteria (see above) were invited to participate. All data with an exception of enhanced participant demographics were collected prospectively.

### Screening procedure

All eligible participants provided signed informed consent before entering the study. Consenting participants were assigned a deidentifying patient ID code, which was then used on all study documentation and recorded ECGs. Participants were asked to complete a basic demographics form (e.g., age, sex, ethnicity, height, weight, current smoking and drinking habits) prior to screening. Screening then followed the process outlined in Fig 1. The radial pulse of the participant was measured for 60 seconds (s), and this was then followed by an $_{SL}$ECG, recorded for 30 s. Only one ECG was recorded, unless the ECG was of poor quality, then a second was recorded. The data of the last ECG recorded for each participant were used for

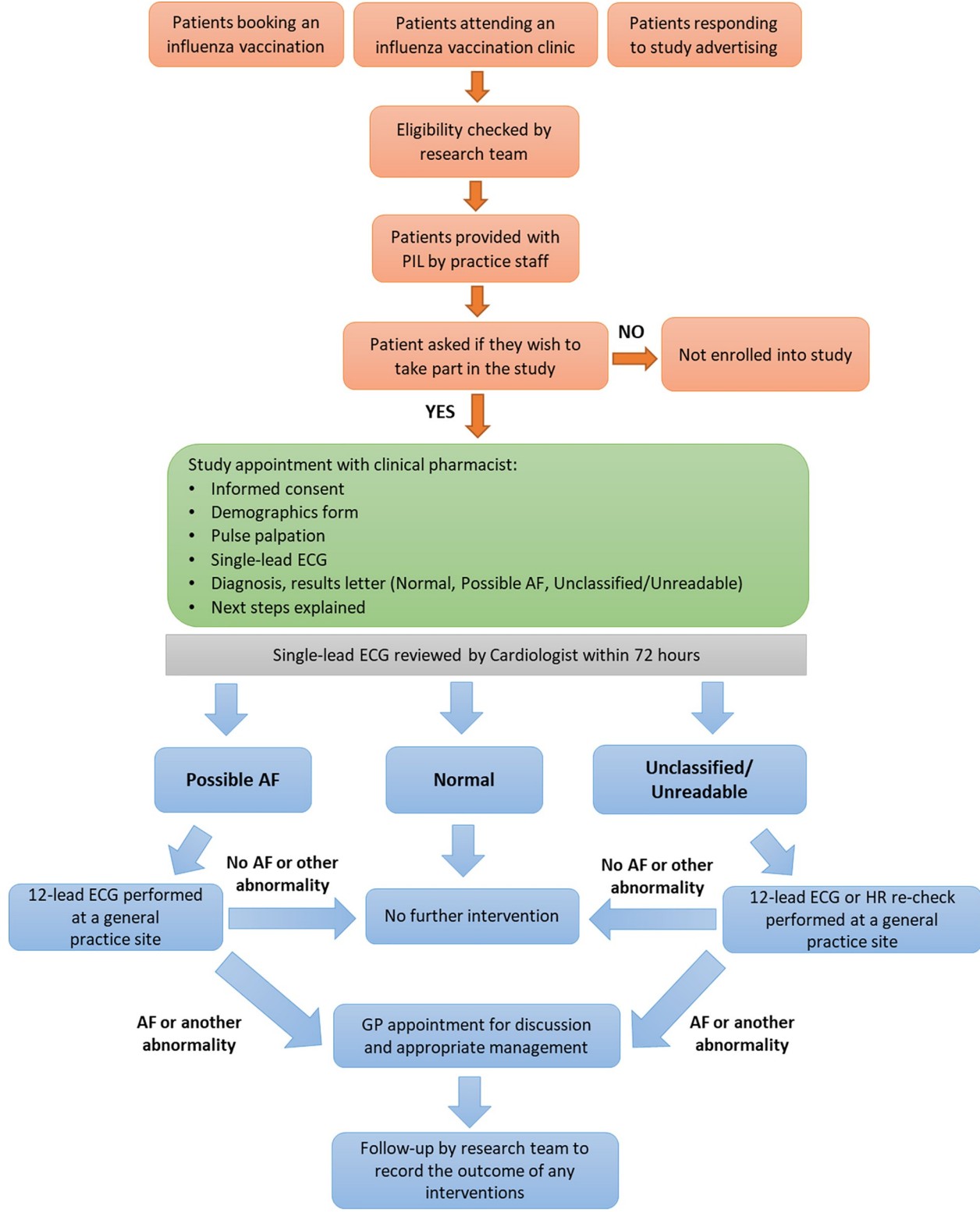

**Fig 1. Pharmacists Detecting Atrial Fibrillation study intervention flowchart.** AF, atrial fibrillation; GP, general practitioner; HR, heart rate; PIL, participant information leaflet.

subsequent analysis. The ECG was assessed and interpreted by the clinical pharmacist. Their assessment of the ECG was explained to the participant and noted, along with the quality of the ECG recording. The clinical pharmacists were not blinded from knowing the provisional diagnosis of the KMD algorithm. All ECGs were emailed to the study cardiologist via the NHS.net email system for overreading. The clinical pharmacist provided the participant with a provisional diagnosis letter that was either 'normal', 'possible AF', 'unclassified' (not sinus rhythm [SR] or AF), or 'unreadable' and advised of the next steps. All ECGs were uploaded to participants' electronic medical record and copies of the consent form and provisional diagnosis letter retained in the study file. The cardiologist's interpretation of the ECG and recommendations for intervention were returned within 72 hours (h). The cardiologist was not blinded from knowing the provisional diagnoses by KMD algorithm or pharmacists and provided pharmacists with regular feedback once each provisional diagnosis was confirmed or rejected. All patient interventions, including confirmatory 12-lead ECGs ($_{12L}$ECGs), were organised by the GP practice in accordance with their normal practice procedures. All participants given either an AF or unclassified/unreadable diagnosis and a recommendation for further intervention by the cardiologist were followed-up by the study team to ensure participants had been offered appropriate treatment from their GP practice. The study team also collated enhanced demographics (e.g., medical history) from all patients with either an AF or unclassified/unreadable diagnosis and a random selection ($n = 100$) of participants who had normal SR at time of screening, for comparison (7/100 participants were excluded from this selection, as their medical records showed that they had either had experienced/were experiencing known AF/paroxysmal AF [PAF]). This sample of participants was selected using the random-cases function of SPSS (v25) in the presence of two researchers.

## Participant experience questionnaire

At the end of the screening appointment, all participants were asked to complete a short, anonymous, patient experience questionnaire consisting of 13 closed and four open questions (S1 Appendix) and were offered the opportunity to take part in future focus groups (reported elsewhere). Completed questionnaires were handed over to the receptionist or posted back to the research team using prepaid envelopes.

## Quantitative data analysis

Apart from the subgroup analysis pertaining to enhanced demographic data of selected participants with suspected 'normal' diagnoses, all data analyses were conducted as prespecified in the study protocol [33]. Continuous variables were reported as a median (interquartile range). Categorical variables, including responses to closed questions of the participant experience questionnaire, were expressed as numbers and percentages (%). The demographics of individuals with and without AF were compared using a Mann-Whitney U test for continuous variables and a Pearson chi-squared or Fisher exact test for categorical variables. Any missing data points were omitted from final analysis, without data imputation. For all statistical comparisons, $p$-values of $<0.05$ were considered significant.

The level of interrater agreement between pulse palpation, pharmacist, device, and cardiologist interpretation of the $_{SL}$ECG was calculated using Cohen's kappa statistic. Diagnostic accuracy measures, including the sensitivity, specificity, percentage agreement with the cardiologist (positive predictive value [PPV]), and the FDR, for each index test were derived from $2 \times 2$ contingency tables using cardiologist's interpretation of $_{SL}$ECG as a reference test. The sensitivity and specificity of the test were defined as its ability to correctly identify those participants with AF (true positives/true positives and false negatives) and without AF (true

negatives/true negatives and false positives), respectively [34]. The overall diagnostic accuracy (correct classification rate) combined these two measures as an assessment of the test's ability to detect both the proportions of true positives and true negatives (true positive and true negatives/total number of participants) [35]. The PPV and FDR were defined as probabilities that the test will identify those with positive diagnoses either correctly (PPV = number of participants who both tested positive and were true positives/total number who tested positive) or incorrectly (FDR = number of participants who both tested positive and were true negatives/total number who tested positive), respectively [34,36].

The diagnostic accuracy of pulse palpation, clinical pharmacist's interpretation, and the device's algorithm (index tests) was compared with the cardiologist's interpretation (reference standard) using a Cochran Q test followed by post hoc McNemar chi-squared tests and a Bonferroni correction for multiple comparisons. Diagnostic accuracy measures were expressed as a mean (95% confidence intervals [CIs]).

Prevalence of new AF diagnoses were determined from the number of confirmed AF-positive $_{12L}$ECGs divided by the total number screened with accompanying 95% CI. False-positive results of each index test were expressed as the number of incorrect AF diagnoses compared with the cardiologist's interpretation, divided by the total number screened (95% CI). All analyses were performed using IBM Statistical Package for Social Sciences (SPSS V.25).

Responses to open-ended questions of participant experience questionnaires were imported into NVivo (V.12) and analysed using content analysis [37], a systematic approach commonly applied to the analysis of verbatim questionnaire data [38]. This included coding the words and frequencies extracted from the questionnaires to identify the frequency of their occurrence and to group them into key themes. The themes were considered alongside responses to closed questions.

## Patient and public involvement

The AF screening protocol and all patient-related information and documents were presented to and scrutinised by the Medway School of Pharmacy, Public Involvement in Pharmacy Studies (PIPS) group prior to submission for ethics approval. Members were also involved in mock training sessions with the clinical pharmacists. The PIPS group comprises interested members of the public. No members of the PIPS group participated in the screening.

The results of the study have been disseminated to participants via various forums including GP practice newsletters, press and media releases (BBC South East, KMTV, and BBC Radio Kent), and social media.

## Ethics

The study was approved by the London-Riverside Research Ethics committee (17/LO/1650) and NHS Health Research Authority. IRAS Project ID is 232663. The study was conducted in accordance with the Medical Research Council's framework for complex interventions [39] and the recommendation for physicians involved in research on human participants adopted by the 18th World Medical Assembly, Helsinki 1964, and later revisions.

## Results

### Participants

A total of 604 participants across four GP practices in Kent underwent a heart rhythm check with a clinical pharmacist. Median age (interquartile range) of the participants was 73 (69–78)

y and 42.7% of participants were male. The majority of participants (96.9%) reported themselves to be White British and had a median body mass index (BMI) of 26.1 (23.5–29.3), Table 1. Nearly 85% of participants only had one $_{SL}$ECG recording (512/604), although two or more ECGs were performed in 15.2% of participants (92/604) in which the first recording was of poor quality.

**Table 1. A summary of participant demographic characteristics ($n$ = 604).**

| Characteristics | $N$ = 604 |
|---|---|
| Age, years | 73 (69–78) |
| Male | 258 (42.7%) |
| White British | 585 (96.9%) |
| White Irish | 3 (0.5%) |
| White American | 2 (0.3%) |
| White Dutch | 2 (0.3%) |
| White other* | 7 (1.2%) |
| Other** | 5 (0.8%) |
| Current alcohol drinker | 380 (62.9%) |
| Alcohol, units/week | 6 (2–14) ($n$ = 372) |
| Current smoker | 54 (8.9%) |
| Height, cm | 167.0 (160.0–174.0) ($n$ = 596) |
| Weight, kg | 73.0 (64.0–83.0) ($n$ = 588) |
| BMI, kg/m$^2$ | 26.1 (23.5–29.3) ($n$ = 585) |
| Heart rate device, bpm | 72 (65–81) |

Continuous variables are expressed as a median (interquartile range). Categorical variables are expressed as a number of participants (% total of the group).

*White European, Flemish, Italian, Scottish, and South African ($n$ = 1 each), and White nonspecified or other ($n$ = 2).

**Kazakh, American, Australian, Hungarian, and Norwegian ($n$ = 1 each).

Abbreviations: BMI, body mass index; bpm, beats per minute (heart rate).

## Screening for AF: Measurement comparison

**Cardiologist.** The cardiologist was able to interpret 99% of the $_{SL}$ECGs recorded, with only 1% (6/604) of the $_{SL}$ECG recorded deemed uninterpretable. From 598/604 $_{SL}$ECGs, the cardiologist diagnosed 503 (83.3%) as normal SR, 26 (4.3%) as possible AF, and 69 (11.4%) as either having unidentifiable or absent P waves or having some other non-AF cardiac abnormality, such as bundle branch block (BBB) and atrioventricular block (AVB) (Fig 2).

**$_{SL}$ECG interpretation by the KMD algorithm.** The KMD algorithm reported 484 (80%) cases as normal SR, 39 (6.5%) cases of possible AF, 75 (12.4%) as unclassified, and six (1.0%) of the ECGs as unreadable. Diagnostic agreement of the algorithm's interpretation of the $_{SL}$ECG compared with the cardiologist's interpretation is illustrated in Fig 2, whereas sensitivity, specificity, and accuracy of diagnosing AF from an $_{SL}$ECG are shown in Table 2. The KMD had a false-positive rate of 2.6% and an FDR of 38.5%.

**$_{SL}$ECG interpretation by the clinical pharmacists.** From the $_{SL}$ECG, clinical pharmacists were asked to record their own interpretation of the $_{SL}$ECG (normal SR, possible AF, unclassified, or unreadable). The clinical pharmacists reported 487 (80.6%) cases as normal SR, 39 (6.5%) cases of possible AF (35 of these matched with the KMD algorithm), 71

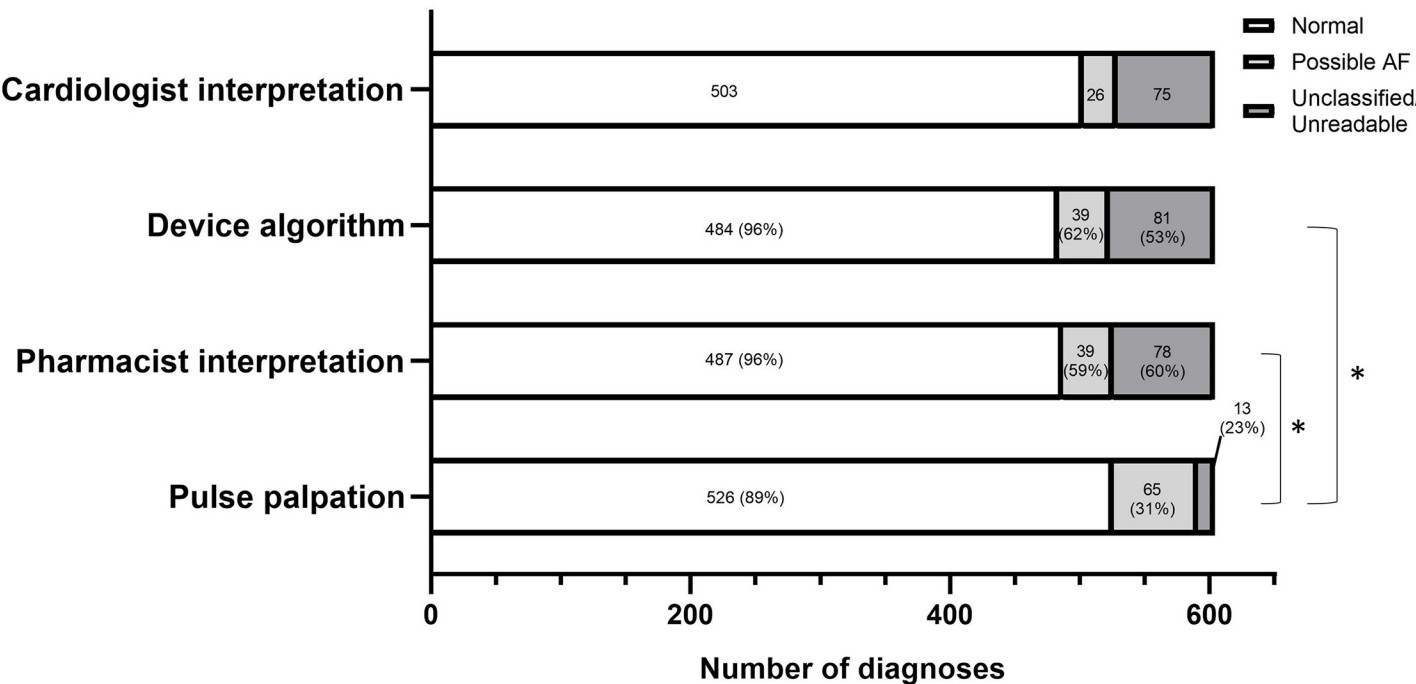

**Fig 2. Screening for AF, measurement comparison.** A breakdown of diagnoses derived from pulse palpation, KMD algorithm, and pharmacist interpretation of the $_{SL}$ECG compared with the cardiologist's interpretation of the $_{SL}$ECG. All data are expressed as the number of cases in each diagnostic category (% diagnostic agreement with cardiologist diagnoses). $^*p = 0.001$ for differences derived from $2 \times 2$ contingency tables for AF-positive and AF-negative diagnoses between KMD and pulse palpation and between pharmacist interpretation and pulse palpation. AF, atrial fibrillation; KMD, Kardia Mobile Device; $_{SL}$ECG, single-lead ECG.

(11.8%) unclassified, and seven (1.2%) of the ECGs as unreadable. Diagnostic agreement of the clinical pharmacist's interpretation of the $_{SL}$ECG compared with the cardiologist's interpretation is illustrated in Fig 2, whereas sensitivity, specificity, and accuracy of diagnosing AF from an $_{SL}$ECG by the clinical pharmacists are shown in Table 2. The clinical pharmacists' interpretation of the $_{SL}$ECG had a false-positive rate of 2.8% and an FDR of 41.0%. The quality of the $_{SL}$ECG recorded for 604 participants was deemed as either excellent (60%), acceptable (33%), poor (5%), or unreadable (2%) by the pharmacists.

**Pulse palpation by the clinical pharmacist.** Heart rate and rhythm interpretation of the pulse was obtained by the pharmacist for 603 participants, with pulse interpretation data missing for one case, in which pulse could not be palpated. Average heart rate was determined by the pharmacist to be 70 beats per minute (bpm) (62–78), compared with 72 bpm (65–81), $n = 604$, obtained using the KMD.

Using pulse palpation alone, pharmacists reported 526 (87.1%) cases as normal SR, 65 (10.8%) cases of possible AF, 12 (2.0%) as unclassified, and one (0.2%) as unreadable (i.e., impalpable). Diagnostic agreement of pulse palpation with the cardiologist's interpretation of the $_{SL}$ECG is illustrated in Fig 2, whereas sensitivity, specificity, and accuracy of diagnosing AF using pulse palpation are shown in Table 2. For pulse interpretation, the false-positive rate and FDRs were high (7.8% and 69.2%, respectively). False-positive AF diagnoses occurred as a consequence of multiple atrial or ventricular ectopic beats ($n = 23$), mild sinus tachycardia ($n = 2$), or bradycardia ($n = 1$), where indicated by the cardiologist's interpretation of the $_{SL}$ECG.

**Table 2. A summary of diagnostic accuracy.** Interpretation of $_{SL}$ECG by the KMD algorithm, pharmacist interpretation, and pulse palpation when compared with the cardiologist interpretation (expressed as a mean (95% CI)).

| Index Test | Sensitivity | Specificity | Accuracy (Correct Classification Rate) | False-Discovery Rate | Cohen's Kappa |
|---|---|---|---|---|---|
| KMD algorithm | 92.3 | 97.4 | 97.2 | 38.5 | 0.72 |
| | (74.9–99.1) | (95.8–98.5) | (95.5–98.4) | (23.4–55.4) | (0.60–0.85) |
| Pharmacist interpretation | 88.5 | 97.2 | 96.9 | 41.0 | 0.69 |
| | (69.9–97.6) | (95.5–98.4) | (95.1–98.1) | (25.6–57.9) | (0.56–0.82) |
| Pulse palpation | 76.9 | 92.2 | 91.6 | 69.2 | 0.40 |
| | (56.4–91.0) | (89.7–94.3) | (89.1–93.7) | (56.6–80.1) | (0.27–0.53) |

Abbreviations: CI, confidence interval; KMD, Kardia Mobile Device; $_{SL}$ECG, single-lead ECG.

## AF prevalence

The total prevalence of 'known' and 'new' AF ascertained by the cardiologist's interpretation of $_{SL}$ECG recordings was 4.3% (26/604). Of these 26 participants, 18/26 (3.0%) had a known medical history of AF, were in AF when screened, and no further action was taken. A total of eight (1.3%) possible-AF participants were referred for a $_{12L}$ECG. Three (0.5%) of these referred participants remained in AF at time of the $_{12L}$ECG. In total, 4/604 (0.7%) participants were diagnosed with 'new' AF as a result of screening after a $_{12L}$ECG confirmation (three with initially suspected 'possible AF' and one with an 'unclassified' diagnosis). Interestingly, of the 18 'known' AF patients, all of whom were receiving oral anticoagulant (OAC) treatment, only seven reported at the time of screening that they experienced AF and were receiving anticoagulation therapy, and three participants were unsure about their diagnosis or treatment, warranting a confirmation in their medical records. All 26 'known' and 'actionable' AF participants were eligible for OACs in accordance with ESC guidelines [8]. Of the 26 participants eligible for OAC therapy, 20 (76.9%) were on OAC therapy at the end of the study (18 with 'known' and two with 'new AF'). An additional participant with a provisional 'unclassified' diagnosis who was diagnosed with 'new' AF following a $_{12L}$ECG was anticoagulated accordingly.

## Demographics of 'new' and 'known' AF participants

Participants with AF were more likely to be male; were significantly older ($p < 0.0001$); had a significantly higher BMI ($p = 0.01$); and a $CHA_2DS_2$-VASc score $\geq 3$ ($p = 0.002$), compared with a random sample ($n = 93$) of participants that were deemed normal SR, at time of screening (Table 3). Extended demographics of participants identified with AF showed that they were significantly more likely to experience hypertension, renal disease, diabetes mellitus, and heart failure (Table 3). Average number of comorbidities per participant from within the AF cohort was 2.0 (1.0–3.0) ($n = 26$), compared with 1.0 (0.0–2.0) for the non-AF cohort ($n = 93$).

## Cost-effectiveness evaluation

The cost-effectiveness of PDAF intervention (see S1 Supporting Information) was estimated with a Markov simulation model built using the cost-utility template by Edlin and colleagues [40], the National Institute for Health and Care Excellence (NICE) costing report for AF [41], and methodology adapted from two previous AF screening studies [42,43]. Cost-effectiveness was evaluated for the KMD and compared with pulse palpation alone or no screening intervention. The intervention was considered to be cost-effective if the estimated incremental cost-effectiveness ratio was under the willingness-to-pay threshold of £20,000/quality-adjusted

**Table 3. A comparison of demographic characteristics between a random sample of participants with normal diagnoses ($n$ = 93) versus those with cardiologist-confirmed AF diagnoses ($n$ = 26).**

|  | Random Sample With Normal Diagnoses ($n$ = 93) | Participants With Cardiologist-Confirmed AF Diagnoses ($n$ = 26) | $p$-Value (Two-Sided) |
|---|---|---|---|
| **Age, years** | 72 (69–76) | 82 (73–85) | <0.0001 |
| **Male** | 36 (38.7) | 15 (57.7) | 0.116 |
| **Current alcohol drinker** | 72 (77.4) | 16 (61.5) | 0.103 |
| **Alcohol, units/week** | 5.5 (2–14) ($n$ = 70) | 10.0 (2–14) ($n$ = 16) | 0.482 |
| **Current smoker** | 6 (6.5) | 3 (11.5) | 0.408 |
| **Height, cm** | 170.0 (162.5–175.0) ($n$ = 91) | 167.5 (162.5–177.5) | 0.634 |
| **Weight, kg** | 73.0 (65.1–81.9) ($n$ = 90) | 78.3 (69.7–97.0) | 0.055 |
| **BMI, kg/m$^2$** | 25.7 (23.1–28.0) ($n$ = 89) | 28.5 (24.2–33.5) | 0.010 |
| **CHA$_2$DS$_2$VASc score** | 3.0 (2.0–3.0) ($n$ = 93) | 3.0 (3.0–4.3) | 0.002 |
| **Hypertension** | 38 (40.9) | 18 (69.2) | 0.010 |
| **Renal disease** | 16 (17.2) | 11 (42.3) | 0.007 |
| **Diabetes mellitus** | 12 (12.9) | 8 (30.8) | 0.041 |
| **Thyroid disease** | 8 (8.6) | 4 (15.4) | 0.293 |
| **Transient ischaemic attack** | 3 (3.2) | 3 (11.5) | 0.117 |
| **Ischaemic heart disease** | 7 (7.5) | 3 (11.5) | 0.454 |
| **Heart failure** | 0 (0.0) | 2 (7.7) | 0.046 |
| **Intracranial bleed** | 1 (1.1) | 1 (3.8) | 0.391 |
| **Peripheral vascular disease** | 4 (4.3) | 0 (0.0) | 0.575 |
| **COPD** | 8 (8.6) | 2 (8.0) | 1.000 |

Continuous variables are expressed as a median (interquartile range). Categorical variables are expressed as a number of participants (% total of the group).

Abbreviations: AF, atrial fibrillation; BMI, body mass index; CHA$_2$DS$_2$-VASc, Congestive heart failure, Hypertension, Age $\geq$ 75, Diabetes, previous Stroke, Vascular disease, Age 65–74 years, Sex category; COPD, chronic obstructive pulmonary disease

life year (QALY) proposed by NICE [44]. At base case assumptions (see S1 Supporting Information), the AF screening strategy was found to be cost-effective in 71.8% and 64.3% of estimates (100,000 simulations run in each case) for KMD and pulse palpation, respectively, compared with no screening intervention. The incremental net benefit compared with no screening strategy was £1,903/patient using the KMD and £946/patient using pulse palpation. If applied to all patients over 65 y old across England and Wales, with 50% uptake of screening and AF newly detected as a result of this screening, this would represent incremental net benefits of around £120 million using the KMD and £50 million using pulse palpation alone.

## Follow-up data and outcomes

Following the initial screening and the cardiologist's interpretation of the $_{SL}$ECG, 87/604 (14.4%) participants with either possible AF or some other cardiac abnormality were referred for a $_{12L}$ECG or heart rate check (Fig 3). The median time between screening and $_{12L}$ECG was 16.0 (11.0–24.0) days (d). One participant declined a $_{12L}$ECG and GP review, and four participants did not respond to an invitation. Of the remaining participants, 28 (4.6%) had normal SR (some identified by GP before $_{12L}$ECG was done); 22 (3.6%) had a previously diagnosed condition and required no further intervention; four had newly diagnosed AF (0.7%), of whom three (0.5%) were initiated on oral anticoagulation; and 28 (4.6%) had a newly diagnosed non-AF cardiovascular condition. Further details concerning the 28 non-AF conditions

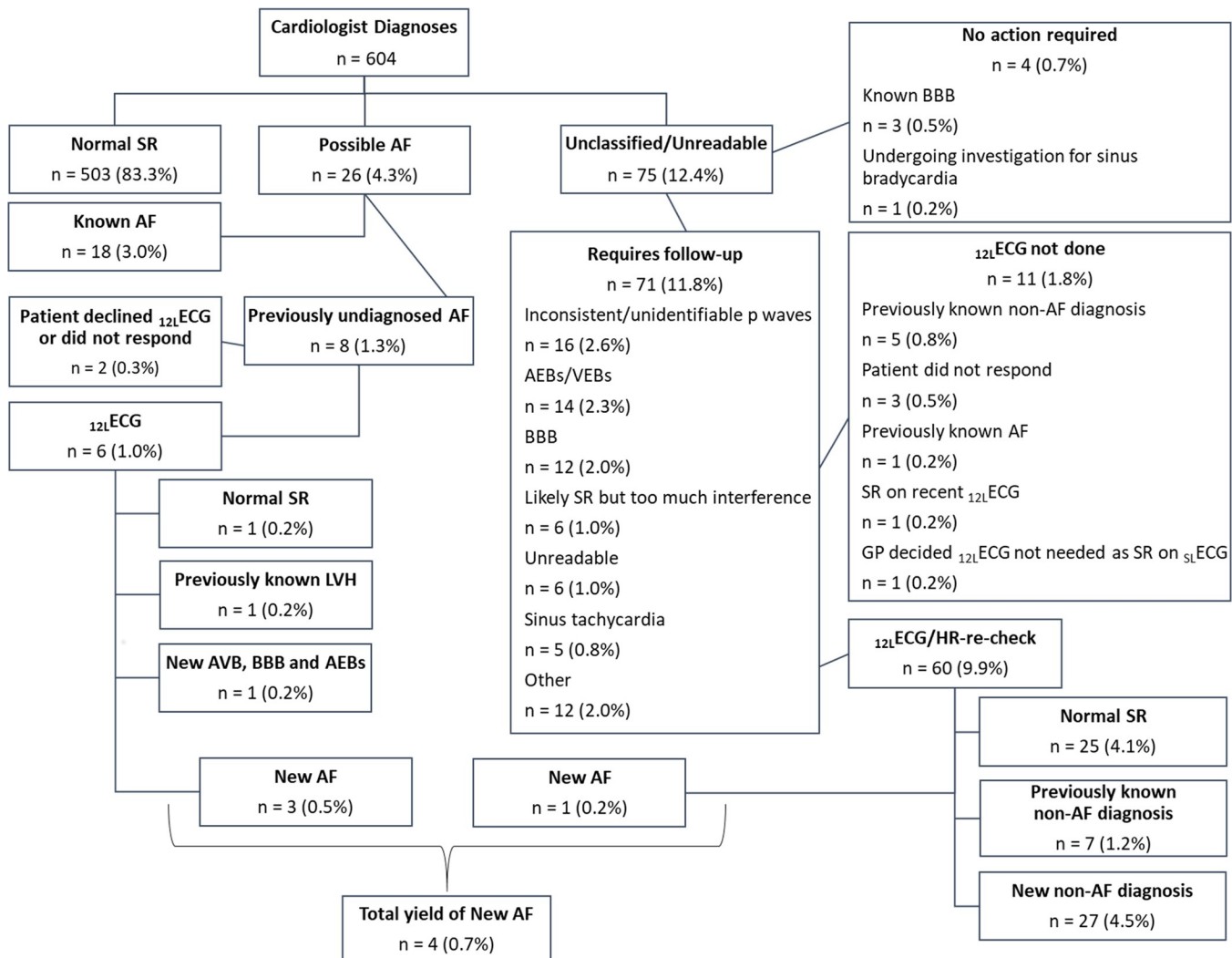

**Fig 3. Flowchart of follow-up actions and outcomes.** $_{12L}$ECG, 12-lead ECG; AEB, atrial ectopic beats; AF, atrial fibrillation; AVB, first-degree atrioventricular block; BBB, bundle branch block; GP, general practitioner; HR, heart rate; LVH, left ventricular hypertrophy; $_{SL}$ECG, single-lead electrocardiogram; SR, sinus rhythm; VEB, ventricular ectopic beats.

identified from the $_{SL}$ECG device and reclassifying of patients following $_{12L}$ECG are shown in Fig 3. None of the participants with a new first-degree AVB (1.3%, 8/604) were referred for pacemaker implantation.

## Participant experience questionnaire

Of the 604 participants screened, 422 (70%) completed a feedback questionnaire. All responding participants rated the overall screening experience as either 'very good' or 'good', and 99% agreed that they would be happy to take part in annual repeat AF screening. Less than half of all respondents (47%) were aware of AF as a condition before they were screened. However, 96% of respondents felt that routine AF screening was either 'very important' or 'important' post screening. In response to open-ended questions, when asked 'Was there anything you particularly liked about the service?', there were 272 recorded comments. Of these, 75 participants praised the 'professional' (14) yet 'relaxed', 'friendly', and 'at ease' nature of the

pharmacist-led screening (61), and 24 stated that the service improved their access to health-care by offering an opportunity to obtain a more rapid provisional diagnosis and reassurance about their health status. Ninety-four participants were particularly pleased with the 'informative' consultation during which they learnt about AF and any information was clearly presented in lay terms they could understand and feel comfortable about, and seven stated that they were particularly impressed with the digital technology that was used. A number of participants (32) particularly liked being able to contribute to clinical research that had a 'preventative medicine' focus. When participants were asked 'Was there anything you particularly disliked about the service?', there were only six comments, with length of time of the appointment noted for two.

## Discussion

### Principal findings

This study produced a number of key findings. Firstly, it showed that clinical pharmacists, assisted by the KMD, were able to detect 24 out of 26 possible AF diagnoses, when compared with the overreading cardiologist. Secondly, participants with confirmed AF had a higher incidence of co- or multimorbidities, including hypertension, renal disease, diabetes, and heart failure. All 'known' and previously 'unknown' AF participants were at risk of stroke and eligible for OACs. Thirdly, screening for AF in the over-65s, combined with another healthcare intervention and using the KMD, was cost-effective and financially beneficial, compared with no screening at all. Fourthly, the participants felt that screening for AF was important, that they were happy for clinical pharmacists to perform the screening, and they were very impressed by the noninvasive digital technology that was used and the information they received from the clinical pharmacist during the appointment. Finally, and arguably the most notable finding, using pulse palpation alone resulted in a larger number of false-positive AF diagnoses compared with the KMD (7.8% vs 2.6%).

### Strengths and limitations of the study

A key strength of this study was that screening was performed during the influenza vaccination season. Since the risk factors associated with AF overlap with those patients invited to participate in the seasonal influenza vaccination [45], combining these health interventions allowed us to optimise recruitment of a relevant and at-risk population of participants with an interest in their own personal well-being and generally in good health at the time of screening. In addition, basing the screening within GP practices and combining it with another healthcare intervention was cost-effective and convenient for patients and ensured that patients received and had access to the necessary follow-on care (e.g., $_{12L}$ECG and treatment) and support for an AF or other cardiovascular diagnosis. This is often missing from studies performed in other primary care settings, such as community pharmacies. However, the space requirement, logistics, and staff endorsement of such a screening strategy in some GP practices may be prohibitive. A single-time-point strategy can also mean that those with PAF are less likely to be detected compared with those with persistent and permanent AF.

Using an $_{SL}$ECG device such as the KMD, which provides a recorded 'snapshot' of a person's heart rhythm, not only was beneficial to patients with possible AF but may also help patients that have other previously undetected cardiovascular complications requiring new treatment or treatment adjustment, for instance, heart failure or sinus bradycardia. However, many of these non-AF cardiovascular diagnoses appeared as an 'unclassified' result and required manual assessment of the ECG by a cardiac specialist and/or confirmation by a $_{12L}$ECG.

A key limitation, although perhaps inevitable of any screening setting, was that the intervention did not reach those patients who either by choice or circumstance were unable to attend the practice for an influenza vaccination. These included the housebound, those with transport issues, patients based in residential and nursing homes, or those that simply do not engage with healthcare interventions such as the influenza vaccination [46–48]. It is likely that many of these patients, who have limited access to healthcare, are perhaps in most need of such screening interventions. Indeed, in studies involving care-home residents, based in the United States and Norway, the AF prevalence was found to range from 6.9% to 18.8% [49,50], which at the peak is eight times higher than the prevalence of AF in the general population [51].

Another limitation of the study was the underrepresentation of UK ethnic minority groups. This study involved predominantly White British participants (97%) and was thus not a true representation of the average UK population, which is 80.5% White British [52] (93.7% in Kent [53]). Expanding this study into areas where there is a higher representation of ethnic minority groups would be required to make it nationally representative.

Measuring the accuracy of the pharmacist to detect AF using the $_{SL}$ECG was limited by the protocol design, in which the automated algorithm was retained on the KMD. The interpretation of the $_{SL}$ECG by the algorithm may have influenced the diagnostic decision made by the pharmacist, potentially increasing the risk of misclassification bias, and thus the level of diagnostic accuracy observed may not be truly representative of their capability. Similarly, the accuracy of the pharmacist to pulse palpate may have been compromised by their relatively limited training and experience; however, this is perhaps representative of the majority of HCPs and certainly the general population for which pulse palpation is being actively promoted.

## Comparison with other studies

The detection of 'new' AF in this study was low (0.7%) but comparable to some of the other studies screening asymptomatic patients ≥65 y old using the KMD where it varied from 0.5% to 1.7% [18,54,55]. This low detection rate may be due in part to the high prevalence (3.6%) of 'known' AF already in this screened population. The overall prevalence of AF 'known' and 'new' in this study was 4.3%, which was consistent with findings from previous studies based in GP and outpatient settings (4.4% [95% CI 4.1–4.6%]), as reviewed by Lowres and colleagues (2013) [51]. Interestingly, enhanced demographics collected from the medical records of 100 random participants that had been determined as having normal SR by the cardiologist showed that 7% of these 100 participants had experienced (or were experiencing) known AF/PAF, although all were in normal SR at time of screening. This suggests that the true prevalence of AF in this population is actually higher than is stated and could potentially be as high as 12.3%, as was found in the STROKESTOP study [56].

Our study showed that those participants with AF were more likely to be older males with a higher BMI and a $CHA_2DS_2$-VASc score not lower than 3.0. These data are consistent with previous studies [11,12,14]. Our data also highlighted that none of the AF cohort had 'lone' or idiopathic AF, but all had one or more conventional risk factors for AF, of which hypertension, renal disease, and diabetes mellitus were the most common [11,13,14]. These findings are consistent with the literature, which has shown 'lone' AF to affect as little as 3% of the AF population and to occur mainly in those with PAF who are under the age of 60 [57], whereas the presence of co- or multimorbidities is much more common in those with AF and reported in a number of studies [12,14,54] and is associated with an increased lifetime risk of developing AF [11]. Targeted AF screening of the older population experiencing one or more risk factors that

overlap with the medical indications recommended for the influenza vaccination [45] would likely make this a viable screening strategy.

Importantly, this study directly compared the accuracy of pulse palpation by the clinical pharmacist with the use of digital technology (KMD). Few studies have directly compared pulse palpation with the newly available digital technology for the detection of AF, despite pulse palpation being the recommended method for first-line detection of AF by NICE and charities such as the Arrhythmia Alliance [31,58]. Three studies reported that pulse palpation had much lower specificity than the newer technology [59–61]. Indirect comparisons reported in systematic reviews demonstrated that pulse palpation in six studies showed reasonable sensitivity (0.92 [0.85–0.96]) as a technique; however, specificity (0.82 [0.76–0.88]) was much lower compared with other methods [29]. In this study, sensitivity and specificity of pulse palpation by the clinical pharmacist was much lower than using the KMD. The KMD had superior specificity in the detection of AF, with over 5%, fewer false-positive results, than pulse palpation. The operating capabilities of the KMD and its algorithm in this study were also found to be comparable to previous studies in similar or different settings, where sensitivity and specificity varied between 55% and 100% and 82% and 99%, respectively [42,62–65].

Reliance on pulse palpation alone would have resulted in a higher number of false positives and false negatives. Interestingly, few studies quote the percentage of false discoveries, i.e., the number of AF diagnoses from all the potential AF diagnoses that were incorrect. This is perhaps because these numbers appear to be alarmingly high. In this study, pulse palpation had an FDR of 69%. In other words, using pulse palpation in isolation would have resulted in 65 out of 604 participants being informed that they potentially had AF, but 45 of these would have been incorrect. The KMD had an FDR of 38.5% (15 out of 39). This is not a negligible amount, but considerably better than for pulse palpation. The issue of using pulse palpation as a first step in the detection of AF is that an irregular pulse is an indicator not just of AF but also of many other conditions [29], and therefore, 70%–87% of all pulse irregularities will not be AF [66]. Consequently, mass screening using pulse palpation will lead to a high number of false positives and, to a lesser extent, false negatives when used solely as a screening test for AF [29,59,60]. For many patients, being told that they possibly had AF would likely cause undue worry and concern if not dealt with correctly by those doing the screening and would be particularly problematic if the patient was independently screening themselves.

This study also demonstrated that screening for AF in primary care during the influenza vaccination season, using a KMD device and clinical pharmacist, was likely to be cost-effective in nearly 72% of cases at a threshold of £20,000 per QALY, compared with no screening at all. These health economic outcomes are aligned with health economic evaluations in other studies using similar conditions [42,43,67].

Interestingly, AF screening results presented here revealed that only seven out 18 patients with previously diagnosed AF were fully aware of their condition. In line with these findings, less than 50% of respondents to questionnaires were aware of AF and related health risks prior to being screened. Although the phenomenon of poor AF awareness amongst the general public is not new [68], it highlights the value of healthcare education provided by qualified HCPs, such as pharmacists, undertaking the screening. In turn, respondents to the questionnaire appreciated the informative and user-friendly consultation with the pharmacist, which improved their access to healthcare and provided immediate reassurance. As reported by previous AF screening studies in primary care [17,69,70], patients were also fascinated by $_{SL}$ECG technology, showcasing the potential of using KMD as a multipurpose screening and educational tool by future AF screening initiatives.

## Conclusion and policy implications

Future screening initiatives will require the involvement of HCPs based in general practices, in particular, clinical pharmacists. Clinical pharmacists can mitigate stress that may occur due to false discoveries; have the potential to treat and manage both the condition and the associated risk factors linked to other coexisting diseases; and can educate the population about the disease. Their participation can assure the longevity of any future AF screening programmes. This study highlights the need for a change in guidelines to move from less reliable and less sensitive practices such as pulse palpation as the first line of AF detection to the adoption of specifically purposed modern technology.

## Future direction

The present study has demonstrated that coupling an AF screening initiative with the influenza vaccination programme is feasible and cost-effective and has a high degree of acceptability to patients. However, key questions remain relating to whether this model could be upscaled and delivered by pharmacists or other HCPs, in all GP practices, and without the 'insurance' of a cardiologist screening all ECGs. Acceptability by patients has been reported here; however, the key barriers perceived by policymakers that have prevented the adoption of a national programme are yet to be explored. Furthermore, the service is not equitable, because although it was freely available to all, those who are more proactive about their health are more likely to participate. Further research, therefore, needs to focus on more inclusive strategies to ensure that routine AF screening is available to those from differing social, economic, and cultural backgrounds.

## Supporting information

**S1 STARD Checklist. STARD, Standard for the reporting of diagnostic accuracy studies checklist.**
(PDF)

**S1 Supporting Information. Cost-effectiveness evaluation** Probabilistic sensitivity analysis, Markov simulation model, Monte Carlo simulation model.
(DOCX)

**S1 Appendix. Patient questionnaire template.**
(PDF)

**S2 Appendix. Demographic, pulse palpation, ECG, case report data.**
(XLSX)

**S3 Appendix. Enhanced demographic and follow-up data.** Medical history, $CHA_2DS_2VASc$ score, HAS-BLED score, investigations, anticoagulation. CHA2DS2-VASc, Congestive heart failure, Hypertension, Age $\geq$ 75 years, Diabetes, previous Stroke, Vascular disease, Age 65–74 years, Sex category.
(XLSX)

**S4 Appendix. Demographic comparison of random normal and possible-AF participants.** Medical history, $CHA_2DS_2VASc$ score, HAS-BLED score. AF, atrial fibrillation; $CHA_2DS_2$-VASc, Congestive heart failure, Hypertension, Age $\geq$ 75 years, Diabetes, previous Stroke, Vascular disease, Age 65–74 years, Sex category.
(XLSX)

**S5 Appendix. Participant feedback questionnaire responses.**
(XLSX)

## Acknowledgments

We thank Kent Community NHS Foundation Trust for providing and allowing clinical pharmacists from their team to participate in this study. We also thank all of the participating GP surgeries from the NHS Canterbury and Coastal Clinical Commissioning Group for supporting this research, with special thanks to the Medical Research Administration Lead, Nichola Lee, who assisted with patient follow-up and data collection. Finally, we would like to thank the members of the PIPS group for all their help and advice.

## Author Contributions

**Conceptualization:** Adrian J. Stewart, Alistair Mathie, Emma L. Veale.

**Data curation:** Vilius Savickas, Alistair Mathie.

**Formal analysis:** Vilius Savickas, Alistair Mathie, Emma L. Veale.

**Funding acquisition:** Sarah A. Corlett, Alistair Mathie, Emma L. Veale.

**Investigation:** Vilius Savickas, Adrian J. Stewart, Melanie Rees-Roberts, Vanessa Short, Alistair Mathie, Emma L. Veale.

**Methodology:** Adrian J. Stewart, Melanie Rees-Roberts, Sukvinder K. Bhamra, Sarah A. Corlett, Alistair Mathie, Emma L. Veale.

**Project administration:** Melanie Rees-Roberts, Vanessa Short, Emma L. Veale.

**Supervision:** Vanessa Short, Sukvinder K. Bhamra, Alistair Mathie, Emma L. Veale.

**Validation:** Adrian J. Stewart, Emma L. Veale.

**Visualization:** Sukvinder K. Bhamra, Sarah A. Corlett, Emma L. Veale.

**Writing – original draft:** Vilius Savickas, Alistair Mathie, Emma L. Veale.

**Writing – review & editing:** Vilius Savickas, Adrian J. Stewart, Melanie Rees-Roberts, Vanessa Short, Sukvinder K. Bhamra, Sarah A. Corlett, Alistair Mathie, Emma L. Veale.

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
