## [Editor Report · Decision Letter 0]

28 Jan 2020

Dear Dr Veale, 

Thank you for submitting your manuscript entitled "Systematic population screening for atrial fibrillation by clinical pharmacists in general practice during the influenza vaccination season: the PDAF study." for consideration by PLOS Medicine.

Your manuscript has now been evaluated by the PLOS Medicine editorial staff and I am writing to let you know that we would like to send your submission out for external peer review.

Kind regards,

Helen Howard, for Clare Stone PhD 

Acting Editor-in-Chief

PLOS Medicine 

plosmedicine.org

---

## [Decision Letter · Decision Letter 1]

17 Apr 2020

Dear Dr. Veale,

Thank you very much for submitting your manuscript "Systematic population screening for atrial fibrillation by clinical pharmacists in general practice during the influenza vaccination season: the PDAF study." (PMEDICINE-D-20-00198R1) for consideration at PLOS Medicine. 

[LINK]

In light of these reviews, I am afraid that we will not be able to accept the manuscript for publication in the journal in its current form, but we would like to consider a revised version that addresses the reviewers' and editors' comments. Obviously we cannot make any decision about publication until we have seen the revised manuscript and your response, and we plan to seek re-review by one or more of the reviewers. 

We expect to receive your revised manuscript by May 08 2020 11:59PM. Please email us (plosmedicine@plos.org) if you have any questions or concerns.

We look forward to receiving your revised manuscript. 

Sincerely,

Artur Arikainen, 

Associate Editor 

PLOS Medicine

plosmedicine.org

1. The reviewers noted that your manuscript is possibly too long – please try to condense your findings more concisely. Additionally, the reviewers raised concerns over the usefulness of the economic evaluation presented in your study. While you do not necessarily need to remove the economic data altogether, I would strongly advise that you present these results in a manner more concise, and more relevant to your main conclusions.

2. Please revise your title according to PLOS Medicine's style. Your title must be nondeclarative and not a question. It should begin with main concept if possible. "Effect of" should be used only if causality can be inferred, i.e., for an RCT. Please place the study design ("A randomized controlled trial," "A retrospective study," "A modelling study," etc.) in the subtitle (ie, after a colon).

3. In your Abstract, please include some overall demographical data for your patients, eg. age, sex.

4. Please ensure that all corresponding p values are provided for all quantitative results in the abstract and throughout your manuscript.

5. In Table 1, we ask that you expand the Ethnicity: “Other” category to show more granularity and to avoid any possible perceptions of marginalisation.

7. Please reiterate a brief description of the content analysis approach cited on line 201.

8. In your STARD checklist, please use section and paragraph numbers, rather than page numbers.

9. PLOS Medicine requires that the de-identified data underlying the specific results in a published article be made available, without restrictions on access, in a public repository or as Supporting Information at the time of article publication, provided it is legal and ethical to do so. Please see the policy at http://journals.plos.org/plosmedicine/s/data-availability and FAQs at http://journals.plos.org/plosmedicine/s/data-availability#loc-faqs-for-data-policy

10. Please provide a copy of the questionnaire used in your study as a Supporting Information file.

11. The usefulness of the word cloud is limited, so we would advise not including it in your manuscript.

Comments from the reviewers:

Reviewer #1: This is a manuscript on screening for atrial fibrillation performed by clinical pharmacists during influenza vaccination season in a general practice setting. Individuals attending influenza vaccination aged above 65 were offered examination using pulse palpation and one-lead ECG (Kardia Mobile). Pharmacists were trained in palpating pulse and in recording and interpreting one-lead ECG. Results from pharmacist examinations were compared to ECG interpretation by cardiologist. The manuscript also includes a health economy assessment and results from feedback given by the participants. The authors concluded that the screening procedure was feasible, economically viable and positively endorsed by participants. 

Comments:

1. In general, I believe that the manuscript is too voluminous, and the present volume will probably discourage some readers from reading the entirety. I suggest putting the health economy data in a separate paper. 

2. The title: If you offer a screening measure to an individual who is attending health care for another reason, the screening is opportunistic, not systematic. The title also implies that this is population screening, but since not all aged above 65 will get the screening invitation, I am reluctant to describe this screening effort as population-based. 

3. Abstract: even though feasibility is the primary outcome, many readers would like to find data on the screening yield as well in the abstract. 

4. I would designate the Kardia ECG device digital technology but hardly novel.

5. It is reported that 604 patients were screened, but how many were offered screening in total and how many declined? This is core data when judging a screening program. Furthermore, if there were data on the number of inhabitants aged over 65 in the catchment areas one could estimate the proportion of the population reached by this screening effort. As a suggestion, the boxes in figure 1 should contain numbers reporting the loss of participants at different stages.

6. Introduction, line 56: Hypertension and heart failure are more common co-morbidities in AF.

7. Clinical pharmacists and cardiologists were not blinded from knowing the ECG interpretation made by the Kardia Mobile algorithm. This could have affected their ECG interpretation and is a major limitation of the validation part of the study and should be recognised as such. 

8. Why was not medical history collected from all participants?

9. This study was intended to validate the use of clinical pharmacists in AF screening and their ECG interpretation as compared to ECG interpretation made by a cardiologist. Since the absolute precision is depending on prevalence of atrial fibrillation, which was low in the screened population, one must consider sample size, i.e. what is the absolute precision in the validation trial with the reported sample size and AF prevalence?

10. Table 1: A percent sign after percentages would increase readability. Abbreviations m and bpm should be explained.

11. It is reported that only 1% of single lead ECG recordings were deemed as uninterpretable by a cardiologist. This is a very low proportion as compared to previous studies. Were there in some cases multiple recordings performed to achieve acceptable signal quality?

12. Paragraph "SLECG interpretation by the clinical pharmacists" starting on line 307: Many of the results in the text are duplicated from table 2, this should be avoided. The Cohens kappa coefficient should be added to table 2. 

13. Paragraph "AF prevalence" starting on line 342: I have difficulties following the figures reported here. It is stated that 18/26 patients had known AF, which implies that the remaining 8 were identified from screening. It is stated on line 345 that 4/604 new cases were diagnosed within the screening project. The figures in the upper part of this paragraph should be revised and clarified. How many participants were diagnosed with new AF?

14. In the same paragraph, it is stated on line 349 that all 26 "actionable" AF participants were eligible for OAC treatment and that 20/26 were on this treatment at the end of the study. How many of these 20 treated patients had known AF and newly diagnosed AF? How many of the patients with known AF (n=18) had OAC treatment on study entry? This figure is vital for the health economy assessment.

15. Line 368: the statistical methods used should only be reported in the methods section.

16. Table 5: Many of the ECG conditions described in table 5 (BBB, low-grade AV-block) will not lead to additional investigations in asymptomatic patients. The therapeutic implication is very different in low-grade and in high-grade AV-block, these should be classified. Were any participants referred for pacemaker implantation?

17. Discussion, line 473: there are no data provided supporting the statement that patients with "possible cardiovascular complications", i.e ECG deviations, have a benefit from the screening procedure. 

18. Discussion, line 507: Given the median age of 73 years of the studied population, I don´t find an AF prevalence of 3.6% high, it is surprisingly low. In the Swedish STROKESTOP 1 trial (Svennberg et al, Circulation 2015:131;2176-84), the baseline prevalence of AF was 9.3% among participants in a population aged 75 and 75. 

19. Discussion, line 525: I don´t think that there is enough data presented to appoint this protocol to the "optimal screening strategy", I suggest removal of this sentence.

20. Conclusion: this paragraph is far too long. I suggest shortening it to 5-10 lines. 

Reviewer #2: Thank you for the opportunity to read and review your manuscript "Systematic population screening for atrial fibrillation by clinical pharmacists in general practice during the influenza vaccination season: the PDAF study". The manuscript is of interest particularly to readers engaged in AF screening, as well as those interested in the involvement of pharmacists in public health initiatives and medical practice.

GENERAL COMMENT

The text is somehow repetitive and written in a verbose style. Much of this could be improved by rephrasing and possibly move some parts to an additional Supplementary document. I.e. details about the participants experience questionnaire; details about how the screening was perceived, which are mentioned in Results and repeated in the Discussion. Furthermore, much results are presented in detail both in the text and in Tables, which may be unnecessary for all numbers.

MAJOR COMMENTS

The manuscript seems long, as it is perceived by this reviewer. A full word count is not available as far as I can see, and can not be retrieved from the available PDF. I trust the Editors may comment on the length of the manuscript with regard to the requirements of the journal, but also for the readability, I would strongly suggest shortening. For example, the Introduction may benefit from shortening, such as the section from page 3, line 64 to page 5, line 92, largely desribes (and discusses) challenges within the UK primary care system. This could be shortened to a few sentences with regard to the focus of this manuscript. 

I think parts of the Methods description, which are very well described in detail, is presented almost like a protocol, and parts of this could, for the benefit of the reader, be moved to a Supplementary document. For example; page 8, line 160-167 (participants experience questionnaire) could be shortened. Instead of details about the questionnaire, 1-2 sentences could remain in the manuscript, and even more could be included in a Supplementary (even consider writing out in full the questions asked).

The first parts of the Results section are clearly presented. However, you may consider shorten down some of the results repeatedly presented both in the text and in Tables; for example all values for sensitivity, specificity etc. also presented in Table 2.

Furthermore, I would question the relevance of the reported heart rate results (page 15, line 323-327). Although a small but significant difference was found, is this clinical relevant? The manual pulse palpation was followed by the KMD, i.e. not performed simultanously, and this could easily explain the observed difference.

I would question the validity/relevance of the reported sl-ECG interpretation by the clinical pharmacists, who have (if I understand correctly) just recorded the KMD algorithm interpretation (SR, AF etc.) before recording their own interpretation, which must potentially be biased by the algorithm, reflected in very comparable numbers. Were the 39 cases of "possible AF" (as interpreted by the KMD) the exact same participants as the 39 cases recorded as "possible AF" by the pharmacists? Or did the pharmacists question the automatic algorithm? I can see that this is again further discussed in the Discussion section (page 24, line 494-501). I would suggest to omit these results, or report this as registered "per protocol", but reduce the importance of this. You may also consider rephrasing this in the Discussion. I would assume the pharmacists must have been biased by the KMD (and not as stated now, page 24, line 496-497; that the KMD "...may have influenced the diagnostic decision made by the pharmacist...".

This reviewer is not capable of fully assessing the cost-effectiveness evaluation of the screening. However, I would humbly state that I find cost-effectiveness assessments without endpoints (stroke and mortality) troublesome. This is a general comment relevant also for other previous publications regarding AF screening. Although extensive cost-effectiveness analyses have been performed, the costs included/saved in the analysis rely on the basic assumption that screen-detected AF has the same clinical course and risk of stroke, as "ordinary" clinically detected AF. Although this is a likely assumption to make, it is yet to be proven. Furthermore, some numbers in the assumptions made for the "base case", such as a KMD-identified new possible AF rate of 1,3%, is not clear to me. The prevalence of screen-detected AF is 0,7%, and I do not find this number in the analysis. However, this may be right. I am also not fully able to assess how the analysis has taken all other findings (non-AF findings) into account; these may, in a real-life setting, produce a substantial cost for the healthcare system, not to mention unnecessary anxiety and repeated examinations. But, my apologies as I do not have the necessary knowledge to fully assess these analyses.

MINOR COMMENTS

Page 3, line 46: Reference #2 does not seem to be the most representative publication to document a growing prevalence of AF. Examples of relevant references could be Ball et al (Int J Cardiol, 2013) or Willams et al (Am J Cardiol, 2017). 

Page 3, line 48-50: Reference #7 is an industry-initiated "white paper". Although it seems balanced and well written, I would suggest you rather use a comparable relevant and independent publication, such as the white paper "Screening for Atrial Fibrillation" by Freedman et al (Circulation, 2016), also providing a strong case to improve early detection and screening for AF.

Page 28, line 577: Language-wise: "year by year" instead of "year on year"?

Reviewer #3: 

This manuscript evaluated an intervention to perform screenings for atrial fibrillation (AF) during visits to a pharmacist for a routine influenza vaccination. The paper assessed both the accuracy of pharmacists assessments using SL-ECG kardia mobile device and the cost effectiveness of a related screening strategy using a simulation informed by the study. Overall, finding new ways to provide screening for common, but serious, health conditions like AF during provision of routine services is a worthwhile goal. However, I have several concerns about the manuscript.

General comments

1. The two main parts of the study (i.e., the assessment of the accuracy of the screening test and the cost effectiveness study) seem very disconnected. As written, I cannot be certain whether this is simply a disconnect in the way the studies are described or in content. 

2. The cost effectiveness analysis was difficult to follow and lacked key details (some specific instances are pointed out in the specific comments below). If this component of the study is to be included, it needs to be fleshed out so that the purpose (e.g., what is being compared) as well as the base and alternative scenarios are clear.

3. The methods section seemed to omit key details related to the assessment of accuracy of the proposed screening approach. Specific instances are referenced in the specific comments below.

4. Restructuring the introduction to highlight the purpose of both components of the study (i.e., the accuracy assessment and the cost effectiveness study) would strengthen the manuscript.

Specific comments

1. Page 5, line 96: should this line read, "we evaluated the use of a single-lead electrocardiogram device *compared with* pulse palpation alone…"?

2. Page 5, line 103: is there a similar checklist or list of standard for reporting results from cost effectiveness studies?

3. Page 6, line 121: were pharmacists aware who had pre-existing AF?

4. Page 6, line 124: "consecutive sampling" should be described in more detail. Were all patients who were eligible offered screening?

5. Page 6, last paragraph: it seems like there is a sentence or 2 missing here to describe the pharmacists reading of the SL-ECG?

6. Page 8, line 172: what is a "non-parametric" variable?

7. Page 8, line 177: When missing data were omitted from the analysis, how was this done? Were participants with any missing data excluded from the entire study or simply excluded from specific analyses where that data point was required?

8. Page 8, last paragraph: I recommend defining the diagnostic accuracy parameters here (e.g., sensitivity, specificity, positive predictive value, and false discovery rate) here. In addition, the term "accuracy" comes up in the results and is not defined here.

9. Page 9, line 193: This paragraph was difficult to follow. The first sentence appears to define "diagnoses of AF", but the definition appears to be for AF prevalence. In addition, this paragraph is the first reference (that I see, apologies if I have missed it) to the 12-lead ECG. What is this and what is its purpose in the study?

10. Page 9, line 201: Here, the manuscript should note the purpose behind analyzing the questionnaire using a content-analysis approach.

11. Page 10, line 210: where does the 1.3% come from in this line? I thought the prevalence of newly identified AF in the study was 0.7%?

12. Page 10, line 217: The description of the hypothetical screening program needs more detail. Moreover, this sentence makes it unclear if the hypothetical population studied included all those who underwent screening or only those who screened positive and were offered oral anticoagulation medications. 

13. Page 10, line 220: What was the purpose of the 3-month cycles? Were these timepoints when outcomes were assessed? This should be clarified.

14. Page 10, line 229: This section of the paper should clearly describe the base cases and alternative scenarios. For example, costs were varied between 50% and 150% of base case, but unclear what these costs are for.

15. Page 11, first paragraph: incremental cost effectiveness ratios should be defined here at first use.

16. Table 1: the variables described in this table should be defined in the Methods section, particularly defining how "alcohol drinker" and "positive smoking status" were defined. For example, does this include ever using alcohol or current use only?

17. Page 15, line 326: is the difference in measured heart rate simply "statistically significant" or is this difference also clinically meaningful? Offering some guidance to readers, here or in the discussion, would be helpful.

18. Page 16, line 351: from this line and the section on follow-up data and outcomes, it appears that there may be a section missing from the methods where follow-up actions should be described.

19. Page 18, line 382: what was the threshold used to defined "cost effective" in this study?

20. Page 18, line 384: It is unclear to me how "participants" were defined here. Was this a cost for all screened individuals, everyone in the population, or some other group?

21. Discussion: some discussion of variability in results between pharmacists would be helpful. Moreover, the discussion would be strengthened by addressing implementation science questions, such as the time required to train pharmacists on the methods needed and whether one would expect accuracy to decline as the intervention was scaled up.

22. Page 27, line 551: should this be "… better than for pulse palpation."?

[LINK]

---

## [Decision Letter · Decision Letter 2]

28 May 2020

Dear Dr. Veale,

Thank you very much for re-submitting your manuscript "Opportunistic screening for atrial fibrillation by clinical pharmacists in general practice during the influenza vaccination season: a cross-sectional feasibility study." (PMEDICINE-D-20-00198R2) for review by PLOS Medicine.

I have discussed the paper with my colleagues and the academic editor and it was also seen again by three reviewers. I am pleased to say that provided the remaining editorial and production issues are dealt with we are planning to accept the paper for publication in the journal.

[LINK]

We look forward to receiving the revised manuscript by Jun 04 2020 11:59PM. 

Sincerely,

Artur Arikainen, 

Associate Editor 

PLOS Medicine

plosmedicine.org

Requests from Editors:

1. Please implement the reviewers’ final comments. However, please note that you can leave in references in the Discussion section, as long as they are relevant in framing your findings in the context of existing literature.

2. Please amend the Title slightly as follows to mention the locale of the study: “Opportunistic screening for atrial fibrillation by clinical pharmacists in UK general practice during the influenza vaccination season: a cross-sectional feasibility study” (no full stop at the end).

3. Lines 16-17: Please update to: “Growing prevalence of atrial fibrillation (AF) in the ageing population, and its associated life-changing health and resource implications, have led to a need to improve its early detection.”

4. Line 31: Please define “CHA2DS2-VASc”.

5. In your abstract and throughout the paper, please quote p values alongside 95% CI, where available. Lines 31-32: We suggest removing the p values. We generally ask that exact p values are quoted, except where p<0.001, but do not favour quoting them in isolation.

6. Line 39, please replace "overwhelmingly" with "generally", or similar.

7. Please adapt the sentence summarizing study limitations (lines 39-40) to quote one additional limitation, e.g., the limited ethnic representation in the study cohort.

8. Please remove the apostrophe from "its" at line 49.

9. Please adapt reference call-outs to the following style: "... health implications [4,5], combined ..." (no space after commas in square brackets).

10. Line 100: Please give exact date ranges.

11. Please remove trademark symbols, e.g. at line 109.

12. Please substitute "sex" for "gender" where appropriate, e.g. at line 128.

13. Lines 234 and 236: Please clarify in the text whether the figures in square brackets show range or another statistic.

14. Table 3: Please give exact p values for all comparisons, including those that are not significant.

15. Table 4: Please replace this with a flowchart, as also recommended by a reviewer. Please also list the abbreviations alphabetically, and include a definition for PR.

16. Lines 387-393: Please provide quantitative data (eg. proportions of patients) to support these statements.

17. Line 397: Please remove the subjective descriptor “comprehensive”.

18. Line 540: Please avoid stating “effective”, as also recommended by a reviewer.

19. Please correct the link for references 1 and 44.

20. Author Summary:

a. Please include the Author Summary in your main manuscript file – after the Abstract, before the Introduction.

b. Bullet point 2: Replace “morbidities” with “conditions” (for clarity to a non-scientific reader).

c. Bullet point 3: Define GP.

-------

Comments from Reviewers:

Reviewer #1: The authors have addressed the majority of my comments on the initial version. 

I still have a few comments to the revision now presented:

1. The manuscript is still very extensive and not just in terms of word count. There are data on AF prevalence and screening yield, comparison between diagnostic modalities, qualitative data and finally data on cost effectiveness. 

2. I still find it difficult to follow all the figures presented in "AF prevalence". A flowchart would be of great help. 

3. The ECG findings and diagnosis presented in table 4 are mostly benign findings that will not have any relevance for asymptomatic patients, i.e ectopics and most conduction abnormalities. I suggest toning down this part of the findings.

4. In "Future direction", it is stated that "The present study has demonstrated that coupling an AF screening initiative with the influenza vaccination programme is effective." The effectiveness of a screening program is defined as its ability to reduce the morbidity and mortality associated to the screened condition, this is not demonstrated with this study and I suggest rephrasing. 

5. Conclusion is still too long should preferably not contain references, rather conclusions based on data from the present study.

6. The low performance of pulse palpation as a screening tool should be emphasized, and the superiority of the digital device over pulse palpation is one of the major results. 

7. The suitability of a GP office as a screening central for AF to my opinion a theoretical construct. As partly mentioned in the introduction, the current staffing and workload in primary care is not compatible with additional tasks like AF screening, at least not beyond what is made by the tremendously committed investigators in local screening initiatives like in this one. There still very little evidence that AF screening could be done in the primary care setting apart from delimited clinical trials, and to what extent can this study change that?

Reviewer #2: The Authors have provided a satisfactory response and the manuscript has been revised accordingly. I would recommend that the manuscript is accepted for publication. 

Reviewer #3: This revision is very responsive to my previous comments. Only a few thoughts remain:

1. Follow-up from previous review: I recommend replacing "non-parametric" variable with "continuous variable". The variable itself cannot be parametric (parametric assumptions are imposed by the analyst). 

2. I still think the paper could be strengthened by discussion of implementation science type questions. While I agree with the authors that additional analyses are unnecessary, the discussion would be strengthened by considering points like "is the time required to train pharmacists on the methods needed worth the improvements noted" and whether the authors expect accuracy to decline as the intervention was scaled up (and moreover, what could be done to prevent this). These questions are important if the work is expected to inform future adoptions of this intervention in other settings. It might be that the authors believe that this approach can be scaled up without loss of accuracy due to important features of the tool, which would be important to note.

3. Based on the methods section, I expected Table 2 to compare sensitivity, specificity, positive predictive value, and false discovery rate. However, it is unclear what measure "accuracy" in Table 2 refers to? This should be clarified with a footnote or renamed.

4. Page 9, line 192: the definitions for PPV and FDR should be clarified. PPV should be (# who both tested positive and were true positives) / (# who tested positive). Similarly the FDR should be (# who both tested positive and were true negatives)/(# who tested positive). Given that the FDR is simply 1-PPV, it may be more informative to report the PPV and the NPV (negative predictive value = (# true negatives who tested negative)/(# who tested negative). But I leave it up to the authors to determine the most useful metrics.

[LINK]

---

## [Editor Report · Decision Letter 3]

16 Jun 2020

Dear Dr Veale, 

On behalf of my colleagues and the academic editor, Dr. Trygve Berge, I am delighted to inform you that your manuscript entitled "Opportunistic screening for atrial fibrillation by clinical pharmacists in UK general practice during the influenza vaccination season: a cross-sectional feasibility study" (PMEDICINE-D-20-00198R3) has been accepted for publication in PLOS Medicine. 

PRODUCTION PROCESS

PRESS

PROFILE INFORMATION

Thank you again for submitting the manuscript to PLOS Medicine. We look forward to publishing it. 

Best wishes, 

Artur Arikainen, 

Associate Editor 

PLOS Medicine

plosmedicine.org